# Health-related quality of life of patients with type 2 diabetes mellitus at a tertiary care hospital in Ethiopia

**Girma Tekle Gebremariam**[1]*, **Selam Biratu**[1], **Metasebia Alemayehu**[1], **Abraham Gebregziabiher Welie**[2], **Kebede Beyene**[3], **Beate Sander**[4,5,6,7‡], **Gebremedhin Beedemariam Gebretekle**[4,5,8‡]

**1** School of Pharmacy, College of Health Sciences, Addis Ababa University, Addis Ababa, Ethiopia, **2** School of Pharmacy, College of Health Sciences, Mekelle University, Mekelle, Ethiopia, **3** School of Pharmacy, The University of Auckland, Auckland, New Zealand, **4** Institute of Health Policy, Management, and Evaluation, University of Toronto, Toronto, Ontario, Canada, **5** Toronto Health Economics and Technology Assessment (THETA) Collaborative, University Health Network, Toronto, Ontario, Canada, **6** ICES, Toronto, Ontario, Canada, **7** Public Health Ontario, Toronto, Ontario, Canada, **8** Centre for Vaccine-Preventable Diseases, Dalla Lana School of Public Health, University of Toronto, Toronto, Canada

‡ BS and GBG are contributed equally to this work and are joint senior authors.
* girma.tekle@aau.edu.et

**Data Availability Statement:** All relevant data are within the manuscript and its Supporting information files.

## Abstract

### Background

Type 2 diabetes mellitus (T2DM) and its treatment impact patients' physical health as well as emotional and social wellbeing. This study aimed to assess health-related quality of life (HRQoL) and associated factors among patients with T2DM at a tertiary care hospital in Ethiopia.

### Methods

A face-to-face cross-sectional survey was conducted among patients with T2DM at Tikur Anbessa Specialized Hospital in Addis Ababa, Ethiopia. We collected data using a validated Amharic version of the 5-level EuroQoL-5 dimensions (EQ-5D-5L) questionnaire. Descriptive statistics were used to present patient characteristics. Kruskal-Wallis and Mann-Whitney U tests were performed to explore differences in the median scores of EQ-5D-5L utility and visual analog scale (EQ-VAS). Multivariable Tobit regression models were used to identify predictors of HRQoL. Utility scores were calculated using disutility weights of the Ethiopian general population. Statistical significance was determined at $p < 0.05$.

### Results

A total of 360 patients with T2DM participated. Mean (SD) age was 64.43(10.61) years. Reported health problems were mostly in the pain/discomfort (67.3%) dimension followed by mobility (60.5%), whereas the usual activities domain (34.1%) was the least health problem being reported. The median (IQR) EQ-5D-5L utility and EQ-VAS scores were 0.95 (0.88–0.96) and 80 (75.0–85.0), respectively. In multivariable Tobit regression models older

**Funding:** This research was supported, in part, by a Canada Research Chair in Economics of Infectious Diseases held by Beate Sander (CRC-950-232429).

**Competing interests:** The authors declare that they have no competing interests.

**Abbreviations: EQ-5D-5L**, 5-level EuroQol-5 dimensions; **EQ-VAS**, EuroQol Visual Analogue Scale; **HRQoL**, Health-related quality of life; **T2DM**, Type two diabetes mellitus.

age, having poor glycemic control, longer duration of diabetes, insulin usage, obesity, and having diabetes-related complications were significant negative predictors of HRQoL.

## Conclusions

Overall, patients with T2DM had lower HRQoL than the general population, which was attributed to being older age, longer duration of diabetes, insulin use, obesity, inadequate glycemic control, and diabetes-related complications. The utility index we generated can be used in future economic evaluations to inform decisions about alternative interventions and resource allocation.

## Introduction

Type 2 diabetes mellitus (T2DM) is a growing public health challenge associated with significant health, social, and economic burden on patients, families, and healthcare systems [1, 2]. According to the International Diabetes Federation (IDF), an estimated 19.4 million adults aged 20–79 years were living with diabetes in the IDF Africa region in 2019, representing a regional prevalence of 3.9% [1]. In Ethiopia, the number of people with diabetes exceeds 2.57 million (5.2%), making it one of the highest prevalence countries in Sub-Saharan Africa [1, 2]. The global epidemic of diabetes is linked to an increasing rate of an aging population, urbanization, unhealthy eating habits, sedentary lifestyle as well as lack of physical activities [3].

HRQoL is a patient-reported outcome measure that evaluates the extent to how diseases, disability, and treatment affects the health status of patients [4]. It encompasses physical, functional, psychosocial, and emotional functioning domains of quality of life [5, 6]. It can provide information about a person's overall health status because it considers both physical and mental health, and their respective impact on HRQoL [7]. Thus, healthcare providers and researchers use self-reported HRQoL measures to evaluate the burden of disease and its treatments in addition to clinical outcomes in patients with diabetes [8, 9]. Moreover, HRQoL is a relevant input to conduct economic evaluations and identify cost-effective interventions that lead to efficient utilization of scarce resources [10].

Demographic factors are an independent determinants of HRQoL in patients with diabetes, particularly aging is strongly negatively associated with HRQoL [10, 11]. In addition, poor glycemic control and diabetes-related complications can lead to a reduction in HRQoL in patients with diabetes [7, 12, 13]. To augment these, many studies demonstrated that patients with diabetes have poor HRQoL compared to people without diabetes [14–17]. Furthermore, studies indicated that duration of diabetes, insulin use which might be associated with pain of multiple injections, are linked with poor HRQoL [9, 18, 19]. Moreover, the presence of comorbidities among patients with T2DM has been linked to a lower HRQoL [12, 20–23].

Several generic and disease-specific tools have been developed for measuring the HRQoL of patients with diabetes [24–27]. The EQ-5D-5L questionnaire is a generic, preference-based multi-attribute utility HRQoL measure validated for use in clinical and economic evaluations. It has the benefit of being able to convert health states into a single summary score called utility. The EQ-5D-5L is being used in many countries for health technology assessment (HTA) such as the National Institute for Health and Care Excellence (NICE) and other European countries' economic evaluations, which inform resource allocation decisions in many jurisdictions [18, 19, 28, 29].

There are no previous studies on HRQoL of patients with T2DM using EQ-5D-5L in Ethiopia; however, a few studies have explored patients' views on the impact of the disease and its treatments using the Short-Form 36 item health survey (SF-36), and the World Health Organization Quality of Life (WHOQOL-BREF) instrument. These studies demonstrate that general health, environmental, psychological, physical, bodily pain and vitality were the most affected dimensions of the HRQoL. Additionally, the findings illustrate that diabetes-related complications, old age, obesity, duration of diabetes, insulin and oral anti-diabetic medication use were the major predictors that require comprehensive intervention strategies to enhance the HRQoL of patients with diabetes. Furthermore, the findings showed that HRQoL remains the most disregarded component in routine clinical practice in Ethiopia [19, 30–32]. Hence, this study aimed to evaluate the HRQoL and associated factors among patients with T2DM at a tertiary teaching hospital in Ethiopia.

## Methods

### Study design and setting

We conducted a face-to-face cross-sectional survey from January to June 2019 among patients with T2DM at an outpatient diabetes clinic of Tikur Anbessa Specialized Hospital (TASH) in Addis Ababa, Ethiopia. The hospital is the largest and oldest tertiary care hospital in Ethiopia. It has 800 beds and serves over half a million patients per year (330 outpatients and 200,000 inpatients, with an average length of stay of 9.3 days). Of these, 6,000 patients attend the diabetic clinic annually.

### Instruments

We used a validated Amharic version of the EQ-5D-5L questionnaire developed by the Euro-Qol Research Foundation [33, 34] (S1 File). The instrument comprises a short descriptive system questionnaire and EQ-VAS. The first part of the EQ-5D-5L involves the patient self-reported component: patients report about their health status in the descriptive system that comprises five dimensions (mobility, self-care, usual activities, pain/discomfort, and anxiety/depression). Under each dimension, there are five levels: no problems, slight problems, moderate problems, severe problems, and extreme problems; which represent the severity of problems for the specific dimension. Participants were asked to choose a level that reflects their current health state for each dimension. The second part of the instrument is the EQ-VAS, an instrument used for subjective assessment of one's current state of health from the patient's perspective [34]. Using this method, each patient self-rated their health status on a vertical scale that ranges from zero (the worst health state) to 100 (the best health state). We also collected sociodemographic (age, gender, marital status, level of education, employment status, average monthly household income, and social history) and clinical data (duration of diabetes diagnosis, previous hospitalization due to diabetes, antidiabetic medications, comorbidities, presence of diabetes-related complications and number of medications).

### Sample size determination, recruitment, and data collection procedure

The sample size was determined using the simple proportion population formula [35], considering a Z-value of 1.96 with a 95% level of confidence and 5% margin of error. To obtain the largest sample size possible, the proportion (P) for sample size estimate was set at 50% of patients with T2DM who rated their overall perceived HRQoL as good. Moreover, sample size adjustment was made since the target populations were less than 10,000. The sample size was calculated to be 360 [35]. We were unable to employ a systematic random sampling strategy

due to the limited time available for the investigation and the small number of patients with T2DM. Instead, study participants were recruited consecutively until we reached the required sample size. Patients were eligible if they were 18 years of age or older and diagnosed with T2DM at least six months before data collection. We excluded patients with gestational diabetes, type 1 diabetes mellitus, and those who had a cognitive/mental problem as per the clinical chart. Data were collected by final-year pharmacy students. All data collectors were trained to ensure uniformity and reduce inter-observer bias in data collection. The purpose and procedure of the study were explained to all study participants before data collection. During the data collection process, data collectors clarified queries raised about the questionnaire by the patients. Information about the health status and sociodemographic characteristics such as gender, age, marital status, occupation, level of education, alcohol, smoking status, lifestyle modifications, and average monthly household income, were collected through face-to-face interviews. Information about the duration of T2DM, history of previous hospitalization, current antidiabetic medications, diabetes-related complications, comorbidities, fasting blood sugar (FBS) level (mg/dl), hemoglobin A1c (HbA1c, %), and the number of medications used were obtained from patients' medical records.

## Ethics

The study was approved by the Ethics Review Board of the School of Pharmacy, Addis Ababa University, Ethiopia *(Protocol#: ERB/SOP/75/04/2019)*. Written informed consent was obtained from all study participants who were able to read and write before data collection. We obtained verbal informed consent for illiterate participants. Most of our illiterate participants had no accompanying family members or relatives; thus, obtaining proxy written consent was not practical. Personal identifiers were not collected, and data were reported in aggregate. The data were stored in password-protected computers, and access to data was restricted to the research team.

## Statistical analysis

Descriptive statistics were used to present the demographic and clinical characteristics of the study participants. Differences in the proportions of reported problems with patients' characteristics were tested using the $\chi^2$ test. As the EQ-5D-5L utility and EQ-VAS scores were non-normally distributed *(Kolmogorov–Smirnov test, p< 0.05)*, we presented median (IQR) scores. The Kruskal-Wallis and Mann-Whitney U tests were used to determine differences in the EQ-5D-5L utility and EQ-VAS scores of participants. To explore the potential predictors of HRQoL, multivariable Tobit regression models were employed. Candidate independent variables were chosen based on previous studies [10, 36] and clinical significance. Patients' EQ-5D-5L utility scores were computed using disutility coefficients obtained from the Ethiopian general population [37]. Statistical significance was determined at $p < 0.05$. All statistical analyses were performed using STATA Version 14.

## Results

### Socio-demographic characteristics of the patients

A total of 360 patients with T2DM were interviewed, but data from eight patients were excluded due to incomplete information. Hence, 352 patients were included in the final analyses. The mean (SD) age of the patients was 64.43 (10.61) years, and the majority (70.7%) were 65 years or older. More than half (55.7%) of the patients were female; 249 (70.7%) were

**Table 1. Sociodemographic characteristics of the patients with T2DM (N = 352).**

| Variables | N (%) |
| --- | --- |
| **Gender** | |
| Male | 156 (44.3) |
| Female | 196 (55.7) |
| **Age categories (in years), Mean (SD)** | 64.43 (10.6) |
| < 60 | 144 (40.9) |
| ≥ 65 | 208 (59.1) |
| **Marital status** | |
| Married | 249 (70.7) |
| Unmarried | 103 (29.3) |
| **Educational status** | |
| Illiterate | 110 (31.3) |
| Primary school | 102 (29.1) |
| Secondary and above | 139 (39.6) |
| **Occupational status** | |
| Employed | 130 (36.9) |
| Non-employed | 222 (63.1) |
| **Average monthly household income (ETB)** | |
| < 14.84 US$ | 95 (27.1) |
| ≥ 14.84 US$ | 256 (72.9) |

* US$ 1 = 40.43 Ethiopian Birr (ETB).

married, 110 (31.3%) were illiterate (i.e., they could not read and/or write), and 95 (27.1%) had a household monthly income of less than US$ 14.84 (Table 1).

## Clinical characteristics of patients

Approximately half (48.1%) of the patients had T2DM for more than 10 years, and a quarter (24.1%) of them had a history of hospital admission due to diabetes. Most (89%) patients reported adopting lifestyle modifications. The majority of patients (77.3%) had one or more comorbidities, with hypertension accounting for the largest proportion (43.8%). More than three-fourth (75.9%) of patients were taking oral antidiabetic agents. The majority (91.2%) of patients had poor glycemic control (FBS ≥126 mg/dl), with an average fasting blood sugar level of 164.11(39.26) mg/dl and an average glycated hemoglobin A1c (HbA1c) value of 7.39%. Most (83%) patients had normal body weight, 191 (54.4%) had diabetes-related complications, and 54 (15.4%) were taking more than five medications (Table 2).

## Distribution of EQ-5D-5L dimensions

Patients' self-reported health status for the five dimensions of EQ-5D-5L is presented in Fig 1. The most frequent health problems were reported for the "pain/discomfort dimension" (67.3%, all levels) followed by "mobility" (60.5%, all levels), while the least was in "usual activities" (34.1%, all levels). The distribution of responses with "no problem" or a "perfect health state" (11111) was reported by 28 (8%) patients in the EQ-5D-5L descriptive dimension, while only 3.7% reported the "best health state" (100) in the EQ-VAS. A significantly higher proportion of any problems reported in EQ-5D-5L descriptive dimensions were observed across patients' characteristics such as gender, marital status, older age, comorbidities, types of antidiabetic medications as well as polypharmacy (S1 Table).

**Table 2. Clinical characteristics of patients with T2DM (N = 352).**

| Variables | N (%) |
|---|---|
| **Time since DM diagnosis** | |
| < 5 years | 73 (20.7) |
| 6–10 years | 109 (31.1) |
| > 10 years | 169 (48.1) |
| **History of hospitalization due to DM** | |
| Never | 267 (75.9) |
| One or more | 85 (24.1) |
| **Adoption of lifestyle modification** | |
| Yes | 327 (93.1) |
| No | 24 (6.80) |
| **Types of lifestyle modification** | |
| Dietary | 216 (66.1) |
| Physical activity | 75 (22.9) |
| Dietary and physical activity | 36 (11.0) |
| **Smoking status** | |
| Yes | 31 (8.80) |
| No | 321 (91.2) |
| **Alcohol habits** | |
| Yes | 157 (44.6) |
| No | 195 (55.4) |
| **Comorbidities** | |
| Yes | 272 (77.3) |
| No | 80 (22.7) |
| **Types of comorbidities** | |
| HTN* | 119 (43.8) |
| HTN + HF* | 83 (30.5) |
| Asthma | 15 (5.5) |
| HTN + asthma | 19 (7.0) |
| HTN + RVI* | 16 (5.9) |
| Others | 20 (7.3) |
| **Antidiabetic medications** | |
| Oral | 261 (75.9) |
| Insulin only | 38 (11.0) |
| Oral + Insulin | 45 (13.1) |
| **FBS level, mean (SD)** | 164.1 (39.3) |
| **HbA1c, mean (%)** | 7.39 (0.476) |
| **Body mass index (BMI), Kg/m2** | |
| Normal body weight | 292 (83.0) |
| Obese | 60 (17.0) |
| **Presence of diabetes-related complication** | |
| Yes | 191 (54.4) |
| No | 160 (45.6) |
| **Number of complications** | |
| < 2 complications | 163 (85.3) |
| ≥ 2 complications | 28 (14.7) |
| **Number of medications used** | |
| < 5 | 297 (84.6) |

(*Continued*)

**Table 2.** (Continued)

| Variables | N (%) |
|---|---|
| ≥ 5 | 54 (15.4) |

*HTN = Hypertension, HF = Heart failure, RVI = HIV/AIDS, FBS = Fasting blood sugar level.

### EQ-5D-5L index and EQ-VAS scores

Overall, the median (IQR) EQ-5D-5L utility and EQ-VAS scores were 0.95 (0.88–0.96) and 80 (75.0–85.0), respectively. The distribution of the EQ-5D-5L utility and EQ-VAS scores were skewed towards 1 and 100, respectively (Fig 2). The median EQ-5D-5L utility scores of patients aged 45 years or younger and those with higher household income (> US$ 14.48) were significantly higher than their counterparts. On the other hand, patients who were living with diabetes for a longer duration and taking more than 5 medications had significantly lower EQ-5D-5L index score compared with patients living for a short period of time and patients taking fewer than 5 antidiabetic medications, respectively. The median EQ-VAS in men was significantly higher in comparison to women (83 versus 75; p = 0.038). In the Kruskal-Wallis

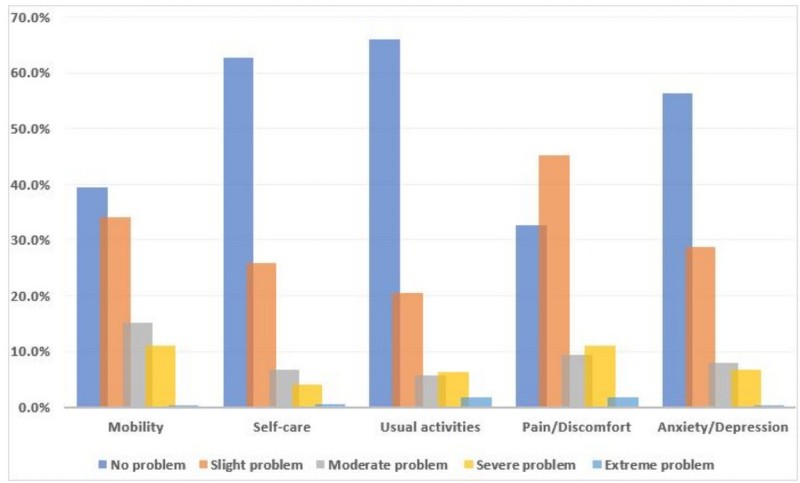

**Fig 1. Percentage distribution of self-reported health problems among patients with T2DM.**

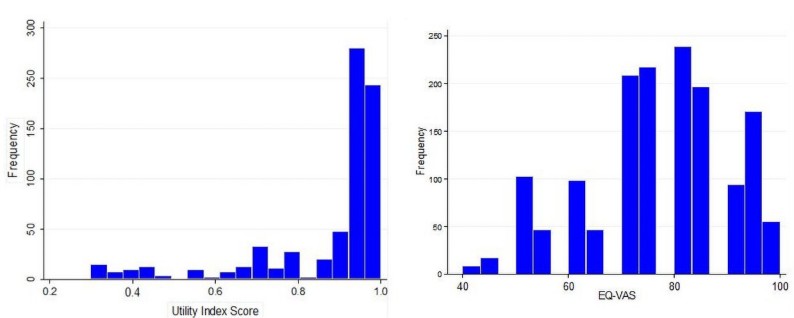

**Fig 2. Distribution of the EQ-5D-5L utility and EQ-VAS scores of patients with T2DM.**

analyses, lower median EQ-5D-5L utility scores were reported in those who were on insulin (0.88), those taking a combination of glibenclamide and insulin (0.44) than metformin alone (0.96) *(p < 0.05)*. Besides, the difference between the median scores of the EQ-5D-5L utility and EQ-VAS scores with controlled and uncontrolled HbA1c was significant *(p < 0.05)* (Table 3).

### Predictors of health-related quality of life

In the multivariable Tobit regression model, having diabetes-related complications (β = -0.029; 95% CI = -0.047; -0.011; *p-value < 0.05*), insulin usage (β = -0.173, 95% CI = -0.227; -0.119, *p-value < 0.05*), being obese (β = -0.071, 95% CI = -0.116; -0.025, *p-value < 0.05*), and longer duration since diabetes diagnosis (β = -0.003, 95% CI = -0.005; -0.001, *p-value < 0.05*) were significant negative predictors of the EQ-5D-5L utility score. Likewise, older age, higher FBS values, and having one or more comorbidities were negatively associated with EQ-5D-5L utility. Whereas higher HbA1c (β = -4.29, *p-value < 0.05*) and smoking (β = -5.12, 95% CI = -9.53; -0.721, *p-value < 0.05*) were significantly negatively associated with the EQ-VAS score. Marital status, occupation, average household income, education level, and polypharmacy were not significantly associated with either the EQ-5D-5L utility or EQ-VAS score. The Tobit regression results are presented in Table 4.

## Discussion

This study aimed to assess the HRQoL and its determinants among patients with T2DM at a tertiary care hospital in Ethiopia, using the EQ-5D-5L instrument. Overall, patients with T2DM reported problems with all descriptive dimensions ranging from 34.1% to 67.3%. We found that the mean health preference-based utility score for patients with T2DM was 0.87, which is lower than the utility score of 0.92 for the general Ethiopian population [37]. Consistent with previous studies [7, 10, 14], duration of diabetes, uncontrolled blood sugar level, insulin usage, obesity, and diabetes-related complications were negatively associated with HRQoL.

Notably, only 8% of patients reported a perfect health state (11111), and approximately 4% reported having the best imaginable health (100%), which is lower than the general population, demonstrating the significant impact of T2DM on patients' HRQoL. Similar to studies conducted in other countries [21, 37, 38], we found that pain/discomfort was the most affected dimension (67.3%) followed by mobility (60.5%). In a Chinese study, however, most of the patients reported problems in pain/discomfort and anxiety/depression [39]. Indeed, these discrepancies among studies could be due to differences in socioeconomic characteristics, duration of diabetes, comorbidities, diabetes-related complications, and health care system-related factors [12, 40]. In line with the previous studies' findings [11, 41], patients with frequent insulin injections and uncontrolled blood sugar levels reported more problems with EQ-5D-5L dimensions. We also found that most participants were physically inactive and should be encouraged to exercise to improve their health status. In this study, usual activities were the least reported problem (34.1%) which is comparable to Iranian patients (32.9%) [42], but lower than in Indonesian patients (48%) [36]. This may indicate that the majority of patients with T2DM are capable of doing daily routine activities such as work and study, and family or leisure activities.

When comparing our findings with previous studies, the mean EQ-5D-5L utility score (0.87) was approximately similar with studies from Iran (0.89) [23], Korea (0.87) [43], Finland (0.85) [44], and Japan (0.86) [45], but higher than reported in an Indonesian study (0.77) [11], and lower than a Chinese study's findings (0.939) [14]. The mean EQ-VAS score was

**Table 3. Median (IQR) differences of EQ-5D-5L utility and EQ-VAS scores with patient demographic and clinical characteristics.**

| Variables | EQ-5D-5L index median (IQR) score | p-value | EQ-VAS median (IQR) score | p-value |
|---|---|---|---|---|
| **Overall scores** | 0.95 (0.88–0.96) | | 80 (75.0–85.0) | |
| **Gender** | | | | |
| Male | 0.95 (0.76–0.97) | 0.198 | 83.0 (70.0–90.0) | **0.038** |
| Female | 0.88 (0.79–0.96) | | 75.0 (70.0–85.0) | |
| **Age category** | | | | |
| < 45 | 0.96 (0.90–0.98) | **0.004** | 80.7 (70.0–95.0) | **0.017** |
| 45–64 | 0.95 (0.91–0.96) | | 80.0 (70.0–90.0) | |
| ≥ 65 | 0.94 (0.75–0.96) | | 75.0 (70.0–85.0) | |
| **Marital status** | | | | |
| Married | 0.95 (0.69–0.96) | 0.708 | 80.0 (60.0–90.0) | 0.529 |
| unmarried | 0.94 (0.89–0.97) | | 80.0 (70.0–85.0) | |
| **Educational status** | | | | |
| Illiterate | 0.86 (0.90–0.96) | **0.013** | 75.0 (60.0–85.0) | 0.070 |
| Primary school | 0.88 (0.93–0.96) | | 80.0 (70.0–85.0) | |
| Secondary school and higher | 0.95 (0.75–0.96) | | 80.0 (70.0–90.0) | |
| **Employment status** | | | | |
| Employed | 0.96 (0.88–0.97) | **0.011** | 80.0 (75.0–90.0) | **0.004** |
| Unemployed | 0.94 (0.79–0.96) | | 75.0 (70.0–85.0) | |
| **Average household income** | | | | |
| < 14.84 US$ | 0.91 (0.78–0.96) | **0.039** | 80.0 (70.0–85.0) | 0.684 |
| ≥ 14.84 US$ | 0.95 (0.90–0.96) | | 75.0 (70.0–85.0) | |
| **Duration of DM** | | | | |
| < 5 years | 0.96 (0.91–0.97) | **0.002** | 87.5 (75.0–95.0) | **0.001** |
| 5–10 years | 0.95 (0.91–0.97) | | 80.0 (70.0–90.0) | |
| > 10 years | 0.93 (0.75–0.96) | | 75.0 (70.0–85.0) | |
| **Hospitalization due to DM** | | | | |
| Never | 0.95 (0.88–0.96) | 0.258 | 80.0 (70.0–85.0) | 0.340 |
| One or more | 0.94 (0.72–0.96) | | 75.0 (60.0–90.0) | |
| **Adoption of lifestyle modification** | | | | |
| Yes | 0.95 (0.88–0.96) | 0.986 | 80.0 (70.0–85.0) | 0.203 |
| No | 0.96 (0.94–0.96) | | 75.0 (70.0–80.0) | |
| **Comorbidities** | | | | |
| Yes | 0.94 (0.77–0.96) | **0.021** | 75.0 (70.0–85.0) | 0.237 |
| No | 0.95 (0.90–0.97) | | 80.0 (70.0–90.0) | |
| **Antidiabetic medications** | | | | |
| Oral | 0.95 (0.91–0.96) | **0.001** | 80.0 (70.0–85.0) | **0.002** |
| Insulin | 0.74 (0.42–0.95) | | 70.0 (53.7–85.0) | |
| Oral + Insulin | 0.95 (0.67–0.97) | | 80.0 (70.0–90.0) | |
| **Number of medications** | | | | |
| < 5 | 0.95 (0.89–0.96) | **0.020** | 80.0 (70.0–90.0) | 0.289 |
| ≥ 5 | 0.93 (0.79–0.96) | | 75.0 (70.0–85.0) | |
| **FBS level** | | | | |
| < 126 mg/dl | 0.95 (0.90–0.96) | **0.009** | 80.0 (70.0–85.0) | 0.186 |
| ≥ 126 mg/dl | 0.93 (0.78–0.95) | | 75.0 (70.0–85.0) | |
| **HbA1c** | | | | |
| < 6.4% | 0.71 (0.71–0.72) | **0.022** | 50.0 (0.00) | **0.006** |
| ≥ 6.4% | 0.95 (0.89–0.96) | | 80.0 (70.0–85.0) | |

(*Continued*)

**Table 3.** (Continued)

| Variables | EQ-5D-5L index median (IQR) score | *p-value* | EQ-VAS median (IQR) score | *p-value* |
|---|---|---|---|---|
| **Body mass index** | | | | |
| Normal | 0.95 (0.91–0.96) | **0.001** | 80.0 (70.0–85.0) | **0.001** |
| Obese | 0.89 (0.72–0.95) | | 75.0 (60.0–80.0) | |
| **Presence of complication** | | | | |
| Yes | 0.94 (0.88–0.97) | **0.001** | 70.0 (65.0–80.0) | **0.001** |
| No | 0.96 (0.88–0.97) | | 80.0 (70.0–90.0) | |

computed to be 76.34 while previous studies reported 56.8 to 80.06% [14, 23, 42, 45, 46]. The differences in socioeconomic characteristics, health system, and patient characteristics, as well as the value set, we used might have contributed to these variations.

Our study showed that patients' HRQoL can be affected by sociodemographic characteristics. The EQ-5D-5L utility score decreased from 0.97 in relatively younger patients with T2DM to 0.85 in older patients, which is consistent with previous studies [9, 47]. This could be

**Table 4. Predictors of HRQoL in patients with Type-2 diabetes mellitus.**

| Variables | EQ-5D-5L utility score | | EQ-VAS score | |
|---|---|---|---|---|
| | β-Coeff. [95% CI] | *p-value* | β-Coeff. [95% CI] | *p-value* |
| **Gender, Female (ref = Male)** | 0.018 [-0.016; 0.053] | 0.686 | 0.455 [-2.42; 3.33] | 0.756 |
| **Level of education (ref = Illiterate)** | | | | |
| Primary | -0.029 [-0.074;0.016] | 0.294 | -0.71 [-4.35;2.92] | 0.699 |
| Secondary and higher | -0.033 [-0.075;0.010] | 0.282 | 0.293 [-3.15;3.73] | 0.867 |
| **Household income (ref = ≤ US$14.84)** | | | | |
| > US$14.84 | 0.035 [-0.003;0.073] | 0.053 | -1.67 [-4.83;1.49] | 0.758 |
| **Age (Ref = < 65 years)** | | | | |
| ≥ 65 years | -0.055 [-0.091; -0.018] | **0.036*** | -2.30 [-5.31;0.691] | 0.143 |
| **Smoking (ref = Non-smoker)** | | | | |
| Smoker | -0.006 [-0.057;0.045] | 0.202 | -5.12 [-9.53; -0.721] | **0.022*** |
| **FBS (mg/dl)** | -0.001 [-0.001; -0.004] | **0.043*** | -0.029 [0.065;0.007] | 0.503 |
| **HbA1c (%)** | -0.003 [-0.039;0.034] | 0.604 | -3.86 [-6.79; -0.935] | **0.012*** |
| **Years since T2DM diagnosis** | -0.003 [-0.005; -0.001] | **0.001*** | -0.313 [0.484;0.142] | **0.001*** |
| **Comorbidity (ref = No)** | | | | |
| Yes | -0.043 [-0.083; -0.003] | 0.582 | -1.74 [-4.97; 1.50] | 0.338 |
| **Antidiabetic medications (Ref = Oral)** | | | | |
| Insulin only | -0.173 [-0.227; -0.119] | **0.002*** | -7.24 [-11.8; -2.71] | **0.002*** |
| Insulin + Oral medications | -0.045 [-0.095;0.005] | 0.360 | 1.45 [-2.81;5.71] | 0.504 |
| **BMI (ref = Normal)** | | | | |
| Obese | -0.071 [-0.116; -0.025] | **0.001*** | -8.09 [-11.82; -4.35] | **0.001*** |
| **Complication (ref = No)** | | | | |
| Yes | -0.029 [-0.047; -0.011] | **0.001*** | -3.19 [-4.66; -1.72] | **0.001*** |
| **Number of medication (Ref = < 5)** | | | | |
| > 5 medications | 0.015 [-0.032;0.028] | 0.884 | 1.87 [-2.23; 5.98] | 0.490 |

* p ≤ 0.05;

**FBS** = Fasting blood sugar level; **HbA1c** = Glycosylated hemoglobin; **DM** = Diabetes Mellitus; **MTF** = Metformin; **BMI** = Body mass index; β-Coeff = Beta coefficient; **CI** = Confidence interval; **SE** = Standard error & **Ref** = Reference.

explained by the progressive increment of different types of comorbidities and diabetes-related complications in the older populations [17, 48]. Consistent with previous studies [20, 23, 41], educational status was correlated with a higher EQ-5D-5L utility score. Patients with better education might have a better understanding of their disease, treatment regimens, and diabetes-related complications. As a result, they could become more diligent about their illness and medication adherence that ultimately enhances their HRQoL [20]. Conversely, poor glycemic control had a negative correlation with EQ-5D-5L utility and EQ-VAS scores; similar findings were reported elsewhere [10, 18]. Javanbakht *et al.* reported that diabetes-related complications such as nephropathy and retinopathy were associated with reduced EQ-5D-5L utility and EQ-VAS scores [43], where the EQ-VAS decrease with the magnitude of 20 in patients who had both complications as compared to no complication, suggesting that having both diabetes-related complications are associated with a marked reduction in HRQoL. Poorly controlled diabetes might increase the risk of disease progression that leads to reduced HRQoL [10, 18].

In Tobit regression models, longer duration of diabetes, uncontrolled blood sugar level, insulin usage, obesity, and presence of diabetic-related complications were negatively associated with EQ-5D-5L utility and EQ-VAS scores [17, 42, 48]. Similarly, Redekop *et al.* and Nguyen *et al.* studies also found insulin therapy, presence of complications, and obesity were associated with lower HRQoL [22, 28]. As demonstrated by *Tran et al.* interventions focused on controlling blood glucose levels, diabetes-related complications, and comorbidities may help to improve HRQoL in patients with diabetes. Similar to our findings, a Vietnamese study demonstrated that having comorbidity reduced the patient's utility [28]. Furthermore, insulin therapy associated decrement in HRQoL might also be explained by the pain of multiple injections. Thus, management protocols for patients with T2DM and clinicians should pay attention to adequately controlling blood glucose levels, diabetes-related complications, normalized body weight, managing comorbidities as well as achieving optimal glycemic control in patients with T2DM to improve HRQoL.

Our study has some limitations. Since this is a cross-sectional study, causality cannot be established between HRQoL and its predictors. As our study was carried out at a single hospital, where the majority of the patients had medical comorbidities and diabetes-related complications, our findings may not represent the health status of diabetes patients across Ethiopia. Third, the current study was conducted over a short time frame and possible changes in the disutility of health problems over time remained unclear. We, therefore recommend longitudinal studies to assess whether differences exist. Despite these limitations, our study generated utility values based on a value set specific for the Ethiopia population, avoiding potential bias given preference-based measures of HRQoL are likely to vary across different populations. Our findings can be used by practitioners and policymakers in designing and implementing strategies aimed at improving diabetes care. Moreover, the study findings can be used as a benchmark to continuously monitor treatment or intervention impact on patients with T2DM. The generic preference-based measures including EQ-5D-5L are the most widely used instruments for assessing health status around the world [33]. However, because they are less sensitive, it is also recommended that disease-specific instruments are included to capture essential aspects of health that are specific to the condition. Ethiopia does not have HTA guidelines, and the choice of preference-based measures remains unclear. This can make it difficult for the ministry to make consistent decisions, given that different preference-based measures yield systematically different values. In many countries, the EQ-5D-5L is the preferred generic preference-based measure for evaluations of health technology [49, 50]. EQ-5D-5L is popular because it is simple to administer, allows for comparisons across interventions and between conditions, and the data reflect the mean value for the population of interest (usually based on

values from members of the general population) [18, 29]. Hence, the utility values we generated using EQ-5D-5L could be used to conduct future cost-utility analysis and prioritize interventions, programs, and policies targeting improving health outcomes of Ethiopian patients with diabetes.

## Conclusions

Our study showed that patients with T2DM had a lower EQ-5D-5L utility than the general population. Patients with T2DM frequently reported problems with pain/discomfort and mobility. Being older, a longer duration of diabetes, insulin use, obesity, inadequate glycemic control, and diabetes-related complications were significant negative predictors of HRQoL. Hence, interventions to improve HRQoL should focus on achieving adequate glycemic control, promoting exercise to reduce obesity, reducing pain/discomfort, and reducing diabetes-related complications. The health preference-based utility value generated in this study could be used to monitor clinical outcomes and conduct economic evaluations of different healthcare interventions in patients with T2DM.

## Supporting information

**S1 Data. All raw data (STATA software).**
(DTA)

**S1 File. English and Amharic version of EQ-5D-5L questionnaires.**
(DOCX)

**S1 Table. Percentage of self-reported health problems among patients with T2DM using EQ-5D-5L descriptive systems.**
(DOCX)

## Acknowledgments

We would like to extend our sincere gratitude to all the study participants, pharmacists, nurses, and physicians working at the study site.

## Author Contributions

**Conceptualization:** Girma Tekle Gebremariam, Selam Biratu, Metasebia Alemayehu, Kebede Beyene, Beate Sander, Gebremedhin Beedemariam Gebretekle.

**Data curation:** Metasebia Alemayehu, Abraham Gebregziabiher Welie, Beate Sander.

**Formal analysis:** Girma Tekle Gebremariam, Selam Biratu, Metasebia Alemayehu, Abraham Gebregziabiher Welie.

**Funding acquisition:** Beate Sander.

**Investigation:** Girma Tekle Gebremariam, Selam Biratu, Metasebia Alemayehu, Abraham Gebregziabiher Welie, Kebede Beyene, Beate Sander, Gebremedhin Beedemariam Gebretekle.

**Methodology:** Girma Tekle Gebremariam, Selam Biratu, Metasebia Alemayehu, Abraham Gebregziabiher Welie, Kebede Beyene, Gebremedhin Beedemariam Gebretekle.

**Project administration:** Kebede Beyene, Beate Sander.

**Resources:** Girma Tekle Gebremariam, Selam Biratu, Metasebia Alemayehu, Abraham Gebregziabiher Welie, Kebede Beyene, Beate Sander, Gebremedhin Beedemariam Gebretekle.

**Software:** Girma Tekle Gebremariam, Selam Biratu, Metasebia Alemayehu, Abraham Gebregziabiher Welie, Kebede Beyene, Gebremedhin Beedemariam Gebretekle.

**Supervision:** Kebede Beyene, Beate Sander, Gebremedhin Beedemariam Gebretekle.

**Visualization:** Abraham Gebregziabiher Welie.

**Writing – original draft:** Girma Tekle Gebremariam, Selam Biratu, Metasebia Alemayehu, Abraham Gebregziabiher Welie.

**Writing – review & editing:** Kebede Beyene, Beate Sander, Gebremedhin Beedemariam Gebretekle.

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
