## [Decision Letter · Decision Letter 0]

23 Jul 2021

PONE-D-21-20037

Health-related quality of life of patients with type 2 diabetes mellitus at a tertiary care hospital in Ethiopia

PLOS ONE

Dear Dr. Gebremariam,

Thank you for submitting your manuscript to PLOS ONE. After careful consideration, we feel that it has merit but does not fully meet PLOS ONE’s publication criteria as it currently stands. Therefore, we invite you to submit a revised version of the manuscript that addresses the points raised during the review process.

I have received the reports from our advisors on your manuscript which you submitted to PLOS ONE.

Based on the comments received, I feel that your manuscript could be reconsidered for publication should you be prepared to incorporate major revisions.

When preparing your revised manuscript, you are asked to carefully consider the reviewer comments below and submit a list of responses to the comments.

Editor Comments: The paper should be checked by a professional speaker of English before complete acceptance.

We look forward to receiving your revised manuscript.

Kind regards,

Muhammad Sajid Hamid Akash

Academic Editor

PLOS ONE

Journal Requirements:

4. We note you have included a table to which you do not refer in the text of your manuscript. Please ensure that you refer to Table 3 in your text; if accepted, production will need this reference to link the reader to the Table.

Reviewers' comments:

Reviewer's Responses to Questions

**Comments to the Author**

1. Is the manuscript technically sound, and do the data support the conclusions?

Reviewer #1: Yes

Reviewer #2: Yes

2. Has the statistical analysis been performed appropriately and rigorously? 

Reviewer #1: Yes

Reviewer #2: Yes

3. Have the authors made all data underlying the findings in their manuscript fully available?

Reviewer #1: No

Reviewer #2: Yes

4. Is the manuscript presented in an intelligible fashion and written in standard English?

Reviewer #1: Yes

Reviewer #2: No

5. Review Comments to the Author

Reviewer #1: Thank you for the invitation to review this manuscript which reports the results of a survey on health-related quality of life of patients with type 2 diabetes mellitus at a tertiary care hospital in Ethiopia. This is much needed and long overdue work, and I commend the authors for conducting this work. The manuscript reads well and has several methodological strengths including the use of a validated Amharic version of EQ-5D-5L instrument, which can be regarded as a good source of evidence for conducting economic evaluations.

My only comment is around the sampling strategy. The authors mentioned that a total of 360 T2DM patients were interviewed. While the recruitment strategy is clear, the sampling techniques and assumptions behind the number “360” is not clear. I also recommend explaining “consecutive sampling method” in a bit more detail here.

Result, analysis, and discussion are well executed and well written.

Reviewer #2: Thank you for the opportunity to review this interesting paper. The study measured health related quality of life and identified associated factors among people with diabetes at a tertiary care hospital in Ethiopia. The study found a median (IQR) EQ-5D-5L utility and EQ-VAS scores of 0.95 (0.88-0.96) and 80 (75.0-85.0), respectively. The two dimensions of EQ-5D-5L for which the most health problems were reported were pain/discomfort (67.3%) and mobility (60.5%). Poor glycemic control, longer duration of diabetes, insulin usage, and being obese had significant negative association with HRQoL.

A very important strength of the study is the use of Ethiopian value sets which makes the findings very suitable for future economic evaluations in the country and other similar settings. However, the authors may want to consider the following points in their revision of the manuscript.

Major comments:

• The methods section provides no information on how sample size was determined and sampling was undertaken. This is important to comment on the appropriateness of inferences even at hospital level.

• While the use of generic instrument in the study is great, there are still benefits of using disease specific tools especially in countries such as Ethiopia where cost utility studies are rare. It would be interesting to consider this perspective in the discussion.

• The introduction is excellent in terms of discussing the factors affecting HRQoL of patients with diabetes. However, it would be great to have a succinct summary of previous HRQoL studies among patients with diabetes in Ethiopia instead of saying “However, there is a paucity of data on diabetic patient’s HRQoL in Ethiopia” even without any reference citation.

Minor comments:

• I am not sure if this is a personal preference but using phrases such as “people with diabetes” or “patients with diabetes” feels much better than “diabetic patients”

• The study hospital has been referred to as “tertiary care hospital” and Specialized Hospital at different parts in the paper. It would be to use one of these consistently throughout.

• In the conclusion of the abstract: I would include one or a couple of the main factor/s instead of leaving it as “several factors”

• According to Table 1, 32% of the respondents were illiterate? Can you please define what “illiterate” mean in this study? And how was the consent signing done for these respondents?

• Table 2 it would be better to present the order of “Yes” and “No” consistently

• Language issues: Although the paper’s message is not compromised due to language issues, there are some grammar issues and typo here and there. Therefore, a careful reading and editing is encouraged. The following are some examples I picked:

o “All data collectors had the training to ensure uniformity and reduce inter-observer bias in data collection.” Should be “All data collectors had a training to ensure uniformity and reduce inter-observer bias in data collection.”

o Page 7 line 160: “More than half (55.7%) of the patients were female; 249 (70.7%)…” should be “More than half (55.7%) of the patients were females; 249 (70.7%)…”. In addition, n(%) reporting has not been consistently followed.

o In Table 1, average monthly household income (ETB), average row there is some erroneously pasted number (2314.87 (2921.9))

o Page 8 line 168: “using lifestyle modifications”… adopting lifestyle modifications and…” The majority (76.4%) had one or more comorbidities”… The majority of patients (76.4%) had one or more comorbidities

o Page 10, line 192: The following sentence needs to state the comparators “On the other hand, patients who were living with diabetes for a longer duration and taking more than 5 medications had significantly lower EQ-5D-5L index and EQ-VAS scores compared to their counterparts with X and Y, respectively

o In Table 2, “Lifestyle modification use” can be more informative if paraphrased as “adoption of lifestyle modification” or similar. Similarly, in the same table, “obesity” should be written as “obese” and “> 2 complication” as “> 2 complications”

o Page 9 line 179: “The most frequent health problems were the pain/discomfort dimension” should be paraphrased as “The most frequent health problems were reported for the “pain/discomfort dimension”

o On page 10 line 189: The distribution of the EQ-5D-5L utility and EQ-VAS scores was were skewed => were skewed…

o Page 16 line 262: “Conversely, glycemic control had a negative correlation with EQ-5D-5L utility and EQ-VAS scores and similar findings were reported elsewhere” is not clear. Was it to mean poorer/weaker glycemic control had a negative correlation with…?

o Page 16 line 267: “suggesting that developing both complications are responsible for a remarkable decline…”

o Page 16 line 274: “I as noted by Tran…”…“The negative association between insulin usaged and…”

6. PLOS authors have the option to publish the peer review history of their article (what does this mean?). If published, this will include your full peer review and any attached files.

Reviewer #1: No

Reviewer #2: **Yes: **Befikadu L. Wubishet

---

## [Author Response · Author response to Decision Letter 0]

17 Sep 2021

Point-by -point response 

Dear Muhammad Sajid Hamid Akash

Academic Editor,

We are pleased to resubmit the revised version of PONE-D-21-20037 ‘‘Health-related quality of life of patients with type 2 diabetes mellitus at a tertiary care hospital in Ethiopia’’ for publication. We appreciate the constructive criticisms of the reviewers and the academic editor. We have addressed their concerns and suggestions as outlined below. All changes in the manuscript are indicated by track changes and detailed explanation as to why the changes are required has been provided here in the point-by-point response. 

We hope that these revisions improve the manuscript, and you will consider it worthy for publication.

Sincerely,

Girma Tekle Gebremariam, Corresponding Author

School of Pharmacy, Addis Ababa University

Zambia Street, Addis Ababa, Ethiopia

Email: girma.tekle@aau.edu.et

Response to reviewers' comments and questions

Academic editor comments and suggestions

Thank you. The manuscript has been revised according to PLOS ONE format.

2. In your Data Availability statement, you have not specified where the minimal data set underlying the results described in your manuscript can be found. PLOS defines a study's minimal data set as the underlying data used to reach the conclusions drawn in the manuscript and any additional data required to replicate the reported study findings in their entirety. 

Thank you for pointing out this. We have mentioned in the manuscripts that ‘‘The raw dataset is available as supporting information.’’ (Please see line 354).

Thank you. The ethics statement has been removed from the last section of the manuscript (Please see line 356).

4. We note you have included a table to which you do not refer in the text of your manuscript. Please ensure that you refer to Table 3 in your text; if accepted, production will need this reference to link the reader to the Table.

Thank you. Table 3 has been cited in the text (please see line 232).

5. Please include captions for your Supporting Information files at the end of your manuscript, and update any in-text citations to match accordingly. 

Thank you. Captions for all supporting information have been included as ‘‘S2 Table: Percentage of self-reported health problems among patients with T2DM using EQ-5D-5L descriptive systems (please see line 519).’’ 

Thank you for your suggestion. We have revised all the reference list and citations. We have also added some additional references (reference # 29-32, 35, 49).

Reviewer #1: 

Thank you for the invitation to review this manuscript which reports the results of a survey on health-related quality of life of patients with type 2 diabetes mellitus at a tertiary care hospital in Ethiopia. This is much needed and long overdue work, and I commend the authors for conducting this work. The manuscript reads well and has several methodological strengths including the use of a validated Amharic version of EQ-5D-5L instrument, which can be regarded as a good source of evidence for conducting economic evaluations.

My only comment is around the sampling strategy. The authors mentioned that a total of 360 T2DM patients were interviewed. While the recruitment strategy is clear, the sampling techniques and assumptions behind the number “360” is not clear. I also recommend explaining “consecutive sampling method” in a bit more detail here.

Response: Thank you for your feedback! In the revised manuscript, we have provided more detailed information on sample size determination and sampling procedure (please see line 133 to 141). In short, we used single proportion with sample size correction formula to determine sample size and consecutive sampling was used to recruit study participants. Consecutive sampling involves recruiting all the people who meet the inclusion criteria and are conveniently available, as part of the sample. It is akin to convenience sampling. In our study, we recruited all eligible T2DM patients consecutively until we reached our pre-determined sample size. We have provided the detail information about sample size determination and the sampling procedure as follows. 

Reviewer #2

Thank you for the opportunity to review this interesting paper. The study measured health related quality of life and identified associated factors among people with diabetes at a tertiary care hospital in Ethiopia. The study found a median (IQR) EQ-5D-5L utility and EQ-VAS scores of 0.95 (0.88-0.96) and 80 (75.0-85.0), respectively. The two dimensions of EQ-5D-5L for which the most health problems were reported were pain/discomfort (67.3%) and mobility (60.5%). Poor glycaemic control, longer duration of diabetes, insulin usage, and being obese had significant negative association with HRQoL. A very important strength of the study is the use of Ethiopian value sets which makes the findings very suitable for future economic evaluations in the country and other similar settings. However, the authors may want to consider the following points in their revision of the manuscript.

Thank you!

Major comments

1. The methods section provides no information on how sample size was determined and sampling was undertaken. This is important to comment on the appropriateness of inferences even at hospital level.

Response: Thank you for your feedback! In the revised manuscript, we have provided more detailed information on sample size determination and sampling procedure (please see line 133 to 141). In short, we used single proportion with sample size correction formula to determine sample size and consecutive sampling was used to recruit study participants. Consecutive sampling involves recruiting all the people who meet the inclusion criteria and are conveniently available, as part of the sample. It is akin to convenience sampling. In our study, we recruited all eligible T2DM patients consecutively until we reached our pre-determined sample size. We have provided the detail information about sample size determination and the sampling procedure as follows.

2. While the use of generic instrument in the study is great, there are still benefits of using disease specific tools especially in countries such as Ethiopia where cost utility studies are rare. It would be interesting to consider this perspective in the discussion.

Response: Thanks for your suggestion. In the last paragraph of the discussion, we added a detail description (please see line 327 to 338). In short, the generic preference-based measures including EQ-5D-5L are the most widely used instruments for assessing health status around the world. However, because they are less sensitive, it is also recommended that disease-specific instruments are included so as to capture essential aspects of health that are specific to the condition. Ethiopia does not have HTA guidelines, and the choice of preference-based measures remains unclear. This can make it difficult for the ministry to make consistent decisions, given that different preference-based measures yield systematically different values. In many countries, the EQ-5D-5L is the preferred generic preference-based measure for evaluations of health technology. EQ-5D-5L is popular because it is simple to administer, allows for comparisons across interventions and between conditions, and the data reflects the mean value for the population of interest.

3. The introduction is excellent in terms of discussing the factors affecting HRQoL of patients with diabetes. However, it would be great to have a succinct summary of previous HRQoL studies among patients with diabetes in Ethiopia instead of saying “However, there is a paucity of data on diabetic patient’s HRQoL in Ethiopia” even without any reference citation.

Thank you for your suggestion, In the last paragraph of the introduction, we added briefly the summary of different findings in Ethiopian among patients with diabetes (please see line 93 to 103). In short, there are no previous studies on HRQoL of patients with T2DM using EQ-5D-5L in Ethiopia; however, a few studies have explored patients' views on the impact of the disease and its treatments. These studies demonstrated that general health, environmental, psychological, physical, bodily pain and vitality were the most affected dimensions of the HRQoL. Additionally, the findings illustrate that diabetes-related complications, old age, obesity, duration of diabetes, insulin and oral anti-diabetic medication use were the major predictors that require comprehensive intervention strategies to enhance the HRQoL of patients with diabetes. Furthermore, the findings showed that HRQoL remains the most disregarded component in routine clinical practice in Ethiopia.

Minor comments

I am not sure if this is a personal preference but using phrases such as “people with diabetes” or “patients with diabetes” feels much better than “diabetic patients”

Thank you for your suggestion. We have used consistently “patients with diabetes”

study hospital has been referred to as “tertiary care hospital” and Specialized Hospital at different parts in the paper. It would be to use one of these consistently throughout.

Thank you for your suggestion. We amended the document and now consistently use the term ‘tertiary care hospitals.’ 

In the conclusion of the abstract: I would include one or a couple of the main factor/s instead of leaving it as “several factors”

Thank you for your suggestion. The major predictors are mentioned in the conclusion part of the abstract section (please see line 52 to 53). In short, we mentioned the major predictors, being older age, longer duration of diabetes, insulin use, obesity, inadequate glycaemic control, and diabetes-related complications.

According to Table 1, 32% of the respondents were illiterate? Can you please define what “illiterate” mean in this study? And how was the consent signing done for these respondents?

Thank you for the feedback. We obtained written consent for those who can read and write but we found it impractical to do the same for the illiterate (who can’t read and/or write) as they are unable to read an informed consent form and understand the risks and benefits of their participation. Besides, most of our illiterate participants had also no accompanying family members so obtaining written proxy consent was not practical. To clarify this, we added some statements in the ‘ethics’ section and reads as follow:” Written informed consent was obtained from all study participants who were able to read and write before data collection. We obtained verbal informed consent for illiterate participants. Most of our illiterate participants had no accompanying family members or relatives; thus, obtaining proxy written consent was not practical (Please see line. 

Table 2 it would be better to present the order of “Yes” and “No” consistently

Thank you. suggestion accepted. We changed and used consistently ‘Yes’ and ‘No’’ (Please see Table 2).

Language issues: Although the paper’s message is not compromised due to language issues, there are some grammar issues and typo here and there. Therefore, a careful reading and editing is encouraged. The following are some examples I picked:

“All data collectors had the training to ensure uniformity and reduce inter-observer bias in data collection.” Should be “All data collectors had a training to ensure uniformity and reduce inter-observer bias in data collection.” 

Thank you. We revised the manuscript to fix issues related to language.

Page 7 line 160: “More than half (55.7%) of the patients were female; 249 (70.7%) …” should be “More than half (55.7%) of the patients were females; 249 (70.7%) …”. In addition, n (%) reporting has not been consistently followed……

Thank you. Suggestion accepted

In Table 1, average monthly household income (ETB), average row there is some erroneously pasted number (2314.87 (2921.9))

 Thank you for pointing out this. Certainly, it is not an error, however our intention to put this number is to know the mean monthly household income. We deleted the number 2314.87 (2921.9) in Table 1 if it confusing to readers. 

Page 8 line 168: “using lifestyle modifications” … adopting lifestyle modifications and…” The majority (76.4%) had one or more comorbidities” … The majority of patients (76.4%) had one or more comorbidities

Thank you. Suggestion accepted (Please see line 193).

Page 10, line 195: The following sentence needs to state the comparators “On the other hand, patients who were living with diabetes for a longer duration and taking more than 5 medications had significantly lower EQ-5D-5L index and EQ-VAS scores compared to their counterparts with X and Y, respectively.

Thank you. Suggestion accepted (Please see line 222-223) ‘On the other hand, patients who were living with diabetes for a longer duration and taking more than 5 medications had significantly lower EQ-5D-5L index and EQ-VAS scores compared to with patients living for a short period of time and patients taking fewer than 5 antidiabetic medications, respectively.’’ 

In Table 2, “Lifestyle modification use” can be more informative if paraphrased as “adoption of lifestyle modification” or similar. Similarly, in the same table, “obesity” should be written as “obese” and “> 2 complication” as “> 2 complications” 

Thank you. All suggestions accepted and please see Table 2.

Page 9 line 182: “The most frequent health problems were the pain/discomfort dimension” should be paraphrased as “The most frequent health problems were reported for the “pain/discomfort dimension” 

Thank you for your suggestion. The sentence rephrased as “The most frequent health problems were reported for the “pain/discomfort dimension” (Please see line 205-206).

On page 10 line 193: The distribution of the EQ-5D-5L utility and EQ-VAS scores was were skewed => were skewed

Thank you. Suggestion accepted

Page 16 line 262: “Conversely, glycaemic control had a negative correlation with EQ-5D-5L utility and EQ-VAS scores and similar findings were reported elsewhere” is not clear. Was it to mean poorer/weaker glycaemic control had a negative correlation with…?

Thank you for your suggestion. The negative correlation refers poor glycaemic control and we amended accordingly to make it with poor glycaemic control (please see line 292).

Page 16 line 267: “suggesting that developing both complications are responsible for a remarkable decline…” 

 Thank you. Suggestion accepted

Page 16 line 274: “I as noted by Tran…” … “The negative association between insulin usaged and…” 

Thank you. Suggestion accepted

Thank you both reviewers!

---

## [Decision Letter · Decision Letter 1]

7 Feb 2022

Health-related quality of life of patients with type 2 diabetes mellitus at a tertiary care hospital in Ethiopia

PONE-D-21-20037R1

Dear Dr. Gebremariam,

We’re pleased to inform you that your manuscript has been judged scientifically suitable for publication and will be formally accepted for publication once it meets all outstanding technical requirements.

Kind regards,

Vijayaprakash Suppiah, PhD

Academic Editor

PLOS ONE

Reviewers' comments:

Reviewer's Responses to Questions

**Comments to the Author**

1. If the authors have adequately addressed your comments raised in a previous round of review and you feel that this manuscript is now acceptable for publication, you may indicate that here to bypass the “Comments to the Author” section, enter your conflict of interest statement in the “Confidential to Editor” section, and submit your "Accept" recommendation.

Reviewer #1: All comments have been addressed

Reviewer #2: All comments have been addressed

2. Is the manuscript technically sound, and do the data support the conclusions?

Reviewer #1: Yes

Reviewer #2: Yes

3. Has the statistical analysis been performed appropriately and rigorously? 

Reviewer #1: Yes

Reviewer #2: Yes

4. Have the authors made all data underlying the findings in their manuscript fully available?

Reviewer #1: No

Reviewer #2: Yes

5. Is the manuscript presented in an intelligible fashion and written in standard English?

Reviewer #1: Yes

Reviewer #2: Yes

6. Review Comments to the Author

Reviewer #1: thank you for revising the manuscript "Health-related quality of life of patients with type 2 diabetes mellitus at a tertiary care hospital in Ethiopia" The comments have now been addressed satisfactorily.

Reviewer #2: Thank you for revising the paper. All the suggestions I had on the first draft have been addressed and I don't have any more suggestions for change.

7. PLOS authors have the option to publish the peer review history of their article (what does this mean?). If published, this will include your full peer review and any attached files.

Reviewer #1: No

Reviewer #2: **Yes: **Befikadu Wubishet

---

## [Editor Report · Acceptance letter]

10 Feb 2022

PONE-D-21-20037R1 

Health-related quality of life of patients with type 2 diabetes mellitus at a tertiary care hospital in Ethiopia 

Dear Dr. Gebremariam:

I'm pleased to inform you that your manuscript has been deemed suitable for publication in PLOS ONE. Congratulations! Your manuscript is now with our production department. 

Kind regards, 

on behalf of

Dr. Vijayaprakash Suppiah 

Academic Editor

PLOS ONE